# Insight into the bacterial communities of the subterranean aphid *Anoecia corni*

**Samir Fakhour**[1,2]**, François Renoz**[2], **Jérôme Ambroise**[3], **Inès Pons**[2], **Christine Noël**[2], **Jean-Luc Gala**[3], **Thierry Hance**[2]

**1** Department of Plant Protection, National Institute for Agricultural Research (INRA), Béni-Mellal, Morocco, **2** Earth and Life Institute, UC Louvain, Louvain-la-Neuve, Belgium, **3** Center for Applied Molecular Technologies (CTMA), Institut de Recherche Expérimentale et Clinique (IREC), UC Louvain, Woluwe-Saint-Lambert, Belgium

⦿ These authors contributed equally to this work.

\* samir.fakhour@inra.ma

**Data Availability Statement:** Data accessibility: European Nucleotide Archive (ENA) accession number of NGS sequence generated for the 16 aphid samples reported in this paper is PRJEB35700.

## Abstract

Many insect species are associated with bacterial partners that can significantly influence their evolutionary ecology. Compared to other insect groups, aphids harbor a bacterial microbiota that has the reputation of being poorly diversified, generally limited to the presence of the obligate nutritional symbiont *Buchnera aphidicola* and some facultative symbionts. In this study, we analyzed the bacterial diversity associated with the dogwood-grass aphid *Anoecia corni*, an aphid species that spends much of its life cycle in a subterranean environment. Little is known about the bacterial diversity associated with aphids displaying such a lifestyle, and one hypothesis is that close contact with the vast microbial community of the rhizosphere could promote the acquisition of a richer bacterial diversity compared to other aphid species. Using 16S rRNA amplicon Illumina sequencing on specimens collected on wheat roots in Morocco, we identified 10 bacterial operational taxonomic units (OTUs) corresponding to five bacterial genera. In addition to the obligate symbiont *Buchnera*, we identified the facultative symbionts *Serratia symbiotica* and *Wolbachia* in certain aphid colonies. The detection of *Wolbachia* is unexpected as it is considered rare in aphids. Moreover, its biological significance remains unknown in these insects. Besides, we also detected *Arsenophonus* and *Dactylopiibacterium carminicum*. These results suggest that, despite its subterranean lifestyle, *A. corni* shelter a bacterial diversity mainly limited to bacterial endosymbionts.

## Introduction

Insects maintain a variety of symbiotic relationships with heritable bacteria that can deeply influence their evolutionary ecology [1–3]. Thanks to their well-studied associations with a wide range of heritable symbiotic bacteria, aphids (Hemiptera: Aphididae) are valuable model systems for studying the evolution of bacterial mutualism in insects [4–6]. Like many insect species that feed on nutrient-deficient diets, aphids typically harbor an ancient nutritional obligate endosymbiont, *Buchnera aphidicola*, confined in specialized cells called bacteriocytes and

**Funding:** This work was supported by the Merit Scholarship Program for High Technology from the Islamic Development Bank [IBD File No. 51/MOR/P33-600029718] and by the Fonds de la Recherche Scientifique (FNRS, FRIA Grant No. 1. E014.17F). The funders had no role in study design and analysis, decision to publish, or preparation of the manuscript.

**Competing interests:** The authors have declared that no competing interests exist.

stably maintained in host populations by vertical transmission [5,7]. In addition to their obligate partner, aphids can also host various facultative endosymbionts that can positively or negatively affect a variety of host phenotypes, depending on the ecological context [4,6].

The functional diversity of facultative symbionts associated with aphids includes γ-proteobacteria [e.g. *Arsenophonus* sp. [8,9], *Regiella insecticola* [10,11], *Serratia symbiotica* [12,13], *Hamiltonella defensa* [14,15], *Rickettsiella viridis* [16,17] and *Candidatus Fukatsuia symbiotica* [18,19]], α-proteobacteria [e.g. the genus *Rickettsia* [20,21] and *Wolbachia* [22,23]] and Mollicutes of *Spiroplasma* genus [24,25]. Ecological effects associated with these bacterial partners include defense against parasites [26–30], body color modification [31,32], heat stress tolerance [33], host plant use and nutrition [34,35] and host reproductive manipulation [36].

In addition to intracellular endosymbiotic bacteria, the aphid microbiota may also include bacterial partners involved in less lasting interactions, which may be transient or even antagonistic, and include gut bacteria, plant associates, pathogens, and environmental contaminants [1,2,37–44]. In aphids, these microorganisms have received limited consideration, notably because of the virtual absence of bacteria in the plant-phloem [45]. However, high-throughput sequencing approaches provide the opportunity to get more complete pictures of the bacterial communities associated with these insects. In this regard, recent studies suggest that the microbiota of some aphid species may be more diverse than previously thought, involving a wider a bacterial diversity that includes members of the genera *Pseudomonas*, *Erwinia*, *Acinetobacter*, *Staphylococcus* and *Pantoea* among others [40,42,46]. Tackling this diversity is crucial to understanding how heritable bacterial endosymbioses are established from free-living lineages. Indeed, recent studies suggest that heritable endosymbionts derive from free-living bacteria, which can sometimes reside in the host plant and in the digestive tract of aphids [39,47–51].

The diversity of the microbiota associated with aphids is linked to their living environment, and the evolutionary acquisition of certain symbionts is probably due to particular habitats. In this context, the soil is an extraordinary reserve of microbiological diversity whose functions are essential to the functioning ecosystems [52]. We therefore hypothesize that aphid species living in close association with the soil, such as the dogwood-grass aphid *Anoecia corni*, are likely to harbor a more diverse and original microbiome than the aphid species strictly present in the areal parts of plants. This particular species notably has access to the xylem tissues of the roots, the first gateway through which many soil-borne bacteria, sometimes pathogenic, transit [53–56].

*A. corni* is a holocylic dioecious species belonging to a genus that includes about twenty aphid species widely distributed in the Holarctic, many whose ecological and taxonomic position remain largely unknown [57,58]. In temperate areas, overwintering eggs hatch on dogwood (*Cornus sanguinea*) during spring, giving rise to a generation of fundatrices. In summer, the alates leave dogwood and migrate onto the roots of grasses and sedges (Poaceae, Cyperaceae) where they are often attended by ants [59]. The microbiota associated with *A. corni* is unknown, and its subterranean lifestyle makes it an ideal candidate to test the hypothesis that close contact with the vast microbial community of the rhizosphere could promote the acquisition of a richer bacterial diversity compared to other aphid species. In this study, we sampled *A. corni* colonies on wheat roots in two regions of Morocco. The 16S rRNA amplicon Illumina sequencing approach was used to examine the composition of the microbiota associated with aphid samples and to clarify the relationship between these two organisms. For this, the evolutionary history was inferred using the Neighbor Joining (NJ). Our results are discussed in light of previous studies on the microbiota associated with other aphid species.

## Materials and methods

### Sample collection and DNA extraction

Apterous adults of *A. corni* were collected during August 2014 on roots of wheat plants (*Triticum* sp.) in Morocco with the kind permission of the landowners. A total of 16 colonies were sampled in two important regions in terms of cereal crops: eight colonies were collected in the locality of Béni Mellal-Khénifra and eight colonies were collected in the locality of Casablanca-Settat (Fig 1; S1 Table in the Supporting Information). Aphid collection consisted of three wingless parthenogenetic adult females per colony, which were immediately immersed in 95% ethanol during collection and preserved at 4°C until use.

### DNA extraction, PCR amplification, library preparation and sequencing

Prior to DNA extractions, aphid samples, each comprising three adult aphids of the same colony, were surface-sterilized with 99% ethanol, 10% bleach and rinsed with sterile water to remove surface contaminants. The genomic DNA was extracted using the DNeasy Blood & Tissue kit (QIAGEN) following the instruction of the manufacturer. The quantity and quality of the DNA extractions were measured with a NanoDrop spectrophotometer (Thermo Scientific, USA). Extractions were then stored at -20°C. After extraction, the genomic 16S rRNA has

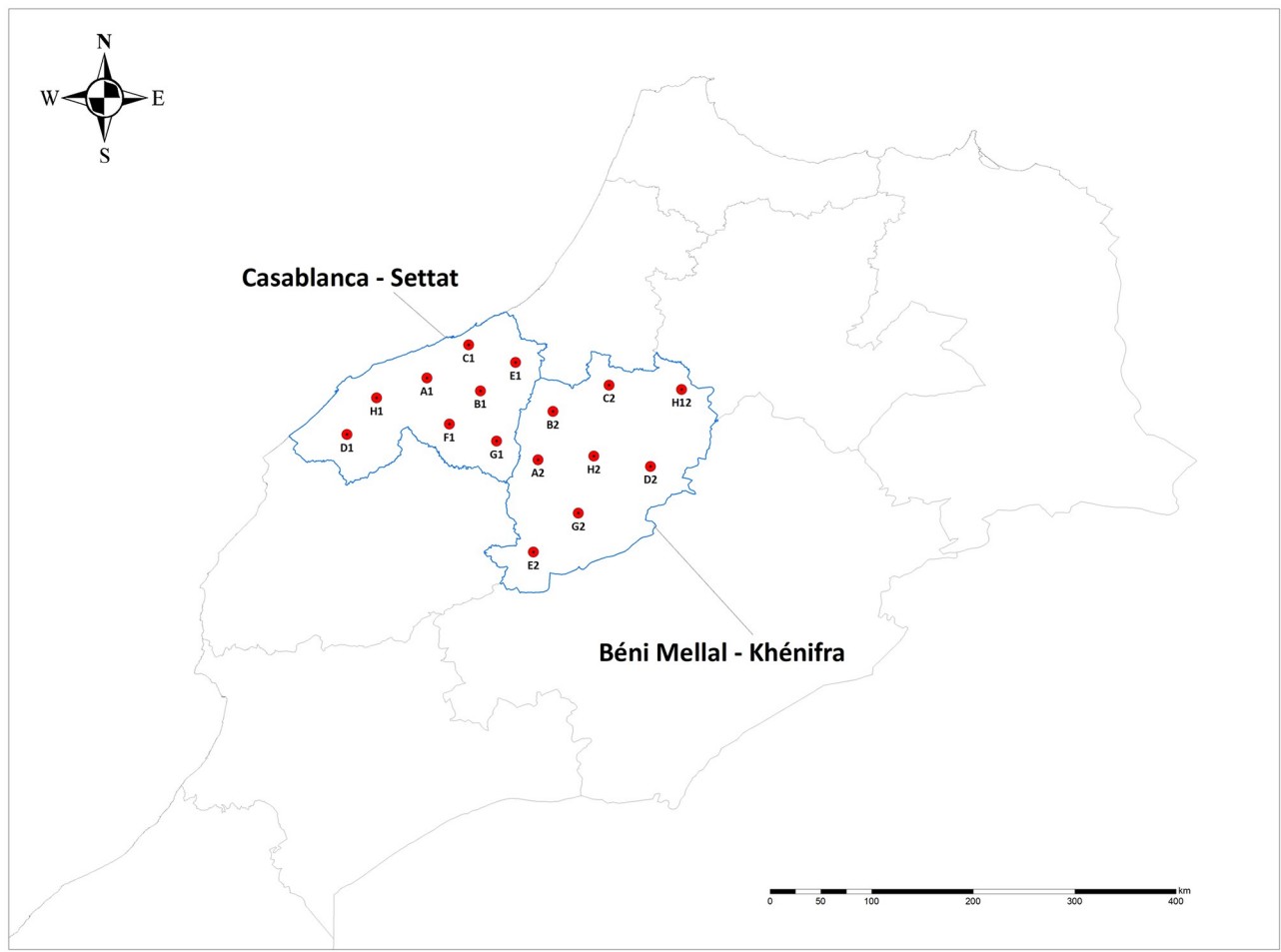

**Fig 1. Geographical location of collection sites of *A. corni* colonies analyzed in this study (for details, see S1 Table in the supporting information).**

been diluted using sterilized ultrapure water in equal concentration (5 ng/μl) from each sample for further steps.

## PCR amplification, library preparation and sequencing

Sequencing libraries were prepared according to the Illumina MiSeq system instructions 16S workflow and as described previously [40]. First, DNA was amplified using universal primers with overhang adapters attached (F: 5' TCGTCGGCAGCGTCAGATGTGTATAAGAGACAGCCTA CGGGNGGCWGCAG and R: 5'GTCTCGTGGGCTCGGAGATGTGTATAAGAGACAGGACTACHVG GGTATCTAATCC) and which targets the V3-V4 variable region of the 16S bacterial rRNA gene. This first PCR step (PCR1) was carried out using a kit 2x KAPA HiFi HotStart ReadyMix in a mixture total volume of 25 μl. The conditions of this first PCR were 95˚C for 3 min (1 cycle); 95˚C for 30 s, 55˚C for 30 s and 72˚C for 30 s (25 cycles), followed by 72˚C for 5 min. PCR products were cleaned with AMPure XP beads. A second-stage PCR (PCR2) was carried out using 5 μl PCR1 purified to attach dual indices and Illumina sequencing adapters using the Nextera XT Index Kit. Different combinations of index (i5 and i7) were used for each sample. The PCR2 was executed under the same conditions of PCR1 but with eight PCR cycles. A clean-up of the PCR2 products with AMPure XP beads was performed before quantification. PCR2 products were quantified and normalized at 7 ng/μl using PicoGreen dsDNA Quantitation Assay and were generated an equimolar pool (7 ng/μl). Before proceeding to high-throughput sequencing (HTS), the final pool was quantified by qPCR (kit KAPA SYBR FAST qPCR ABI Prism readymix KK4604) and 7 pM of denatured final pool was loaded on MiSeq reagent kits v3 (600 cycles). All the processes were carried out by the GIGA-Research Center of the University of Liège (ULiège, Belgium) using Illumina MiSeq Technology for paired-end sequencing (2 × 300 bp reads).

## Data analysis

The data analysis was achieved as previously described [40]. The 16S rRNA Illumina Miseq sequencing reads were analyzed using UPARSE [60] bioinformatics pipeline. For each aphid sample, forward and reverse sequences from paired-end reads were assembled and the resulting consensus sequences were filtered based on their respective quality (expected number of errors <1.0 and a length >450 nt). Sequences with ≥ 97% similarity were assigned to the same operational taxonomic units (OTUs). Chimera were removed using a reference-based filtering with UCHIME and the gold database of the corresponding software. A second level of quality-filtering was carried out in order to discard OTUs with a number of sequences <0.005% of the total number of sequences, as recommended previously [61]. As previously recommended [62,63], an additional filtering was performed by analyzing the negative controls in order to remove OTUs corresponding to potential contaminations. Bacterial taxonomic assignments of each OTU were obtained using the dada2 R package [64] and the Greengenes (v.13.8) database [65]. Finally, to improve the taxonomic assignment, the representative OTUs were compared to the sequences in the GenBank using BLASTn. To compare samples, the number of sequences was standardized or rarefied to 50,000 per sample. After rarefaction, the OTU table was analyzed (See S2 Table. Commands and options used to build the OUT table). All bacterial sequences found in *A. corni* are given in S3 and S4 Tables (Supporting information). Raw data were deposited European Nucleotide Archive (ENA) as a file under accession number PRJEB35700.

Phylogenetic relationships of bacteria associated with *A. corni* and representative endosymbionts of other aphids were established using SeaView v4.6.1 to align 16S rRNA sequences [66] and GBlocks v0.91b [67] to remove poorly aligned positions and divergent regions of DNA

alignments. We selected the best fit evolutionary models using PartitionFinder v1.1.0 [68]. The phylogenetic tree was reconstructed using the neighbor joining method with SeaView v4.6.1 and bootstrap values were computed for each branch node (N = 1000).

## Results

### Library basic statistics

On average, NGS sequencing produced 187,592 bacterial 16S rRNA reads per sample (Table 1).

The assembly of paired sequences resulted in consensus sequences with an average length of 466 bp. The quality-based filtering of the consensus sequences resulted in an average number of high-quality sequences per sample of 150,605.

### OTU clustering and taxonomic assignment

Initially, high-quality reads were clustered using >97% sequence similarity into 23 OTUs. Based on the analysis of the negative controls, 13 OTUs that count for 0.23% of the total number of reads were identified as contaminants and removed (S4 Table). Reads were therefore clustered into 10 biologically relevant OTUs (Table 2 and S3 Table).

All OTUs correspond to the group of Proteobacteria and include three bacterial orders: *Enterobacteriales*, *Rhodocyclales* and *Rickettsiales*. Our results indicate that the microbial profile of *A. corni* is dominated by the order of *Enterobacteriales*, which includes the obligate symbiont *B. aphidicola* and the facultative symbionts *S. symbiotica* and *Arsenophonus*.

*B. aphidicola* was detected in all the samples (100%) and was represented by 5 OTUs (OTUs 1, 2, 3, 7 and 10) that account for 96.23% of all reads. OTUs 1, 7 and 10 differed by only 3 to 4 bp whereas OTUs 2 and 3 differed from these OTUs by 20 to 38 bp. Different *B. aphidicola* haplotypes are present in an aphid colony with the dominance of a single haplotype (Fig 2). The vast majority of the reads clustered into a single OTU for most aphid species (OTUs 1, 2 or 3) and many minor OTUs (OTUs 7 and 10) were detected in all samples. In aphid colonies from the Casablanca-Settat region, The OTU1 (with related minor OTUs 2, 3, 7 and 10) was detected and matched a sequence of *B. aphidicola* previously reported on *Geoica urticularia*. OTUs 2 and 3 were common to all samples from Béni Mellal-Kénifra region and OTU3 was detected on the *Anoecia* genus.

The next most abundant OTUs were presented by the facultative symbiont *S. symbiotica* (OTUs 4 and 5) that account for 3.52% of all reads. Taxonomic identification of bacterial OTUs resulted in three additional taxa including *Wolbachia* (OTU 6), *Arsenophonus* (OTU 8) and *D. carminicum* (OTU 9). Phylogenetic analyses including the symbionts associated with

**Table 1. Summary of sequencing data.**

| *Raw data* | |
| --- | --- |
| Average size (Mb) per sample | 3 |
| Raw number of sequence per sample | 187,592 |
| *After assembly of paired sequences* | |
| Average size (Mb) per sample | 169.4 |
| Average number of sequence per sample | 173,117 |
| Average of the median sequence length | 466 |
| *After quality filtering* | |
| Average size (Mb) per sample | 72.7 |
| Average number of sequence per sample | 150,605 |
| Sequence length (min; median; max) | (450; 465; 584) |

**Table 2. Taxonomic assignment of OTUs by Greengenes and GenBank, including the three top BLAST hits, GenBank accession numbers and % identity.**

| OTU no. | PC reads. | Greengenes identification | Id% | GenBank identification | Accession | Id% |
|---|---|---|---|---|---|---|
| | | Taxon | | Three closest GenBank matches | | |
| OTU_01 | 58.77 | *Buchnera* | 98.28 | *Buchnera aphidicola/Geoica urticularia* | AJ296751.1 | 98.28 |
| | | | | *Buchnera aphidicola/Myzus persicae* | CP002703.1 | 96.34 |
| | | | | *Buchnera aphidicola/Myzus persicae* | CP002701.1 | 96.34 |
| OTU_02 | 20.08 | *Buchnera* | 96.15 | *Buchnera aphidicola/Pemphigus matsumurai* | KF311221.1 | 96.15 |
| | | | | *Buchnera aphidicola/Pemphigus sinobursarius* | KF311219.1 | 96.15 |
| | | | | *Buchnera aphidicola/Pemphigus yunnanensis* | HQ792326.1 | 96.15 |
| OTU_03 | 17.34 | *Buchnera* | 91.24 | *Buchnera aphidicola/Anoecia oenotherae* | CP033012.1 | 97.85 |
| | | | | *Buchnera aphidicola/Anoecia fulviabdominalis* | JX998094.1 | 97.2 |
| | | | | *Buchnera aphidicola/Eulachnus mediterraneus* | LT600356.1 | 94.22 |
| OTU_04 | 2.62 | *Serratia* | 99.14 | *Serratia symbiotica/Aphis fabae* | KT176010.1 | 99.35 |
| | | | | *Serratia symbiotica/soil* | MG287131.1 | 99.14 |
| | | | | *Serratia symbiotica/soil* | KX900450.1 | 99.14 |
| OTU_05 | 0.89 | - | 90.15 | *Serratia symbiotica/Prociphilus longianus* | MG831336.1 | 99.35 |
| | | | | *Serratia symbiotica/Prociphilus longianus* | MG835393.1 | 98.92 |
| | | | | *Serratia symbiotica/Prociphilus longianus* | MG835392.1 | 98.92 |
| OTU_06 | 0.04 | *Wolbachia* | 96.64 | *Wolbachia pipientis/Pentalonia nigronervosa* | KJ786950.1 | 96.64 |
| | | | | *Wolbachia pipientis/Pentalonia nigronervosa* | KJ786949.1 | 96.64 |
| | | | | *Wolbachia pipientis/Pentalonia nigronervosa* | KC522606.1 | 96.64 |
| OTU_7 | 0.01 | *Buchnera* | 98.21 | *Buchnera aphidicola/Geoica urticularia* | AJ296751.1 | 98.21 |
| | | | | *Buchnera aphidicola/Myzus persicae* | CP002703.1 | 95.92 |
| | | | | *Buchnera aphidicola/Myzus persicae* | CP002701.1 | 95.92 |
| OTU_8 | 0.01 | *Candidatus Phlomobacter* | 95.05 | *Arsenophonus/Aleurodicus dispersus* | AY264664.1 | 95.91 |
| | | | | *Arsenophonus/Macrosteles sexnotatus* | AB795344.1 | 95.27 |
| | | | | *Arsenophonus/Stomaphis takahashii* | FJ655541.1 | 95.27 |
| OTU_9 | 0.01 | *Uliginosibacterium* | 98.92 | *Dactylopiibacterium carminicum/Dactylopius opuntiae* | GQ853370.1 | 98.92 |
| | | | | *Dactylopiibacterium carminicum/Dactylopius opuntiae* | GQ853369.1 | 98.92 |
| | | | | *Sphingomonas/soil* | JX944513.2 | 96.34 |
| OTU_10 | 0.01 | *Buchnera* | 98.06 | *Buchnera aphidicola/Geoica urticularia* | AJ296751.1 | 98.06 |
| | | | | *Buchnera aphidicola/Myzus persicae* | CP002703.1 | 96.13 |
| | | | | *Buchnera aphidicola/Myzus persicae* | CP002701.1 | 96.13 |
| OTUs 11–23 | 0.23 | Contaminants identified from negative control analysis | | | | |

PC reads, cluster size in percent; Id, identity %.

*A. corni* (i.e. *B. aphidicola*, *S. symbiotica*, *Arsenophonus* and *Wolbachia*) and representative symbionts of other species of aphids and insects are shown in S1–S4 Figs (Supporting information).

## Diversity of bacterial communities in the samples

The bacterial communities of the samples were mainly composed of the obligate symbiont *B. aphidicola* and mostly complemented by the facultative symbionts *S. symbiotica* (with a high abundance of reads from three samples) and *Wolbachia* (with reads detected from four samples) (Fig 2).

The facultative symbiont *Arsenophonus* was detected in most of the samples (13/16), but with an extremely low number of reads. *D. carminicum* was also found in most of the samples with an extremely low number of reads.

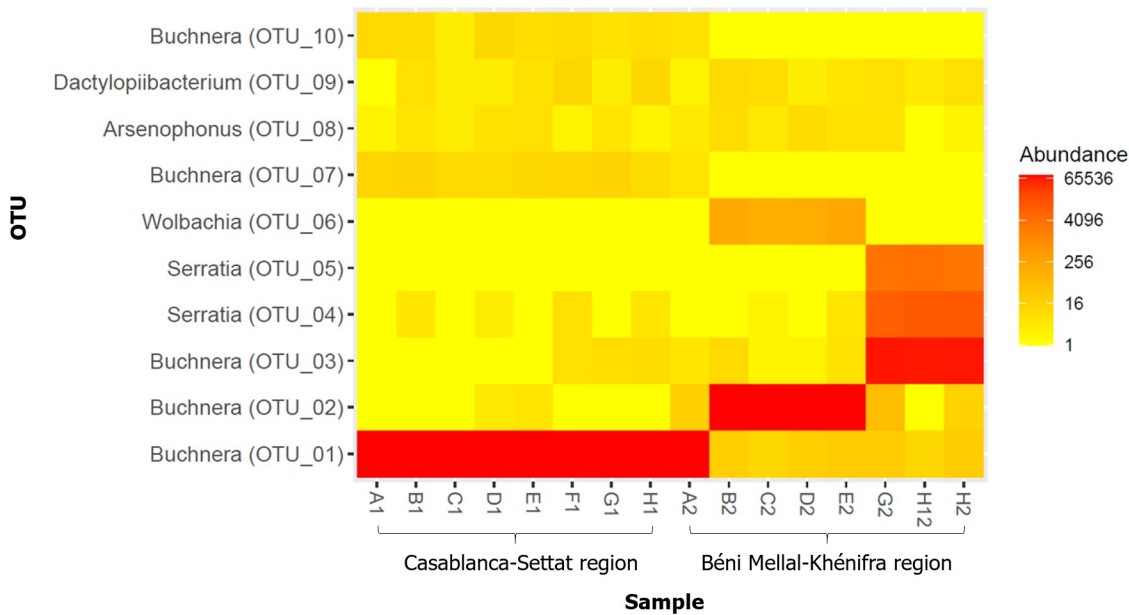

**Fig 2. Relative abundance of bacterial taxa from Illumina sequencing of 16S rRNA amplicons, represented as a heat map based on the log-transformed values.** The warm colors indicating higher and cold colors indicating lower abundance. Each color bar corresponding to one sampled colony.

## Discussion

Our approach based on 16S rRNA enabled us to identify five bacterial genera in the dogwood aphid *A. corni*, a species that spends most of its life cycle on Poaceae roots. Due to its subterranean lifestyle, we expected to find in this species a bacterial diversity that includes environmental bacteria such as gut bacteria, plant associates, pathogens and environmental contaminants. Xylem vessels are the primary entry routes for many soil-borne bacteria infecting plants [56]. Although primarily phloem-feeders, aphids are also capable of ingestion from the xylem vessels, a strategy displayed for maintaining water balance [69,70]. OTUs that were identified correspond to five bacterial genera, mostly related to symbionts already found associated with aphids: *Buchnera*, *Serratia*, *Wolbachia*, *Arsenophonus* and *Dactylopiibacterium*. All these genera have been previously described as symbiotic partners of insects. We did not find any bacterial partners that can be considered as environmental-related (e.g. *Pseudomonas* spp., *Erwinia* spp., etc.) as in the case of other aphid species, including those that feed on cereal crops [40,42].

The nutritional obligate symbiont *Buchnera* was found in all samples with distinctive 16S haplotypes in a single aphid colony. These results might be a consequence of *Buchnera* polyploidy, as evidenced by the 16S rRNA copy-number variation. Alternatively, a clone from a single colony may contain *Buchnera* strains with different haplotypes. Co-infection with multiple *B. aphidicola* strains was reported in several aphid genera [71,72].

The secondary endosymbiont *S. symbiotica* is one of the most common symbiont species in aphid populations [73] and was identified in three of the eight colonies surveyed. This symbiont includes a wide variety of strains ranging from co-obligate nutritional partners, that are mainly found in the Lachninae and the Chaitophorinae subfamilies [12,74], to facultative strains whose reported associated effects in the pea aphid *Acyrthosiphon pisum* are heat stress resistance and protection against parasitoids [75–77]. Strains detected in this study are probably of facultative nature, as *S. symbiotica* was not found in all colonies.

Interestingly, *Wolbachia* was detected in four colonies. This symbiont, an α-proteobacterium, is commonly found in insects and studies suggest that *Wolbachia* is present in at least 65% of arthropod species [78]. It is known to manipulate the reproduction of their host [79], promote the oogenesis in certain wasp species [80], display a nutritional function in certain bedbug and whiteflies species by producing B vitamins [81,82], and is associated with antiviral protection by influencing the vector competence of several species of mosquitoes for viruses [83,84]. Although some studies have reported the presence of *Wolbachia* in aphid populations, it is considered rare in these insects [85–90]. The biological significance of *Wolbachia* in aphids is still unknown. It has been hypothesized that the symbiont play a role in the proliferation of asexual lineages [87], and its role in the production of B vitamins in the banana aphid *Pentalonia nigronervosa* is currently debated [86,91,92]. *Wolbachia* infections in aphids could also be acquired by horizontal transmission from other insects such as parasitoid wasps, known to be infected by this symbiont [93,94]. To our knowledge, the stability of *Wolbachia* infections in aphids has never be tested and *A. corni* could be a suitable candidate to elucidate the biological significance of this symbiont in aphids.

The genera *Arsenophonus* and *Dactylopiibacterium* were also detected in most of the sampled aphids, but with a much lower read abundance than for *S. symbiotica* and *Wolbachia*. *Arsenophonus* is a bacteria found in many insect species including aphids, scale insects, leafhoppers, whiteflies and wasps [8,82,95–97]. Despite the fact that the prevalence of *Arsenophonus* can reach up to 70% in species of the *Aphis* genus [98], the phenotypes associated with this symbiont remain unclear in aphids. Bacteriophages required for protective symbiosis were found in various strains of the symbiont [99], but no defensive properties were found in *Aphis glycines* infected by *Arsenophonus* [98]. Recent studies suggest that *Arsenophonus* may be involved in host nutrition, probably by mediating host plant range [35,100–102]. In whiteflies, comparative genomics suggests that *Arsenophonus* is a source of B vitamins [82]. However, no genome of strains associated with aphids has yet been sequenced. While the presence of a symbiont in specialized host cells such as bacteriocytes and sheath cells are important clues for determining the mutualistic and heritable nature of a symbiont [4], no such information are available for this particular symbiont.

One OTU was assigned to *D. carminicum* (β-proteobacteria, family Rhodocyclaceae). So far, this bacterial species has only been reported in the scale insect species *Dactylopius coccus* (Hemiptera: Coccoidea: Dactylopiidae), where it has been described as a nitrogen-fixing symbiont [103,104]. *Dactylopius coccus* is now well established in Morocco where it ravages the plants of *Opuntia ficus-indica*. Although *D. carminicum* is considered a symbiont capable of passing through the reproductive organs in scale insects, this species remains largely undocumented. It cannot be excluded at present that this newly discovered species resides in the soil, in the host plant or lives in other insects.

In recent years, several studies have characterized overall all the bacteria present in aphids by deep sequencing of 16S rRNA [40,42,45]. A common point throughout these studies is the reduced abundance of environmental bacteria relative to the primary and secondary endosymbionts. Besides theses heritable symbionts and in contrast to our results, many environmental bacteria have been reported in *R. padi*, e.g. the phytopathogenic members of the Pseudomonas genus and some saprophytes of plant and soil (*Acinetobacter* and *Staphyloccocus* genera). Moreover, gut symbiotic bacteria of aphids were also found, i.e. *Pantoea* and *Erwinia* genera [40,42].

In conclusion, although *A. corni* lives in the rhizosphere, an environment that is very rich in bacteria and other microorganisms, the number of bacterial taxa detected in this species is surprisingly low [105]. Despite its subterranean lifestyle, *A. corni* shelter a bacterial diversity mainly limited to known bacterial endosymbionts. Few species of facultative endosymbionts

have been detected in the context of this study. It is now recognized that some species of aphids are more likely to harbor facultative symbionts than others [4,73,106]. It should be noted that our sampling covers only a small part of Morocco, and the diversity of symbionts of aphid populations can change dramatically in response to various environmental conditions [4]. However, our study provides a snapshot of the bacterial community associated with a poorly studied aphid species, and identified bacterial taxa that may play a role in the biology of *A. corni*, in particular the facultative symbionts *Arsenophonus* and *Wolbachia* whose associated phenotypes in aphids are still elusive. *A. corni* could represent a suitable species to investigate the role of these symbiotic bacteria in aphids. Finally, insect-associated bacterial communities, and in particular heritable symbionts, have received much attention in recent decades, somewhat overshadowing the diversity of other types of microorganisms that can associate with insects, such as fungi. For example, in some species of cicadas, grasshoppers, and aphids, certain fungi species have become obligate symbionts by replacing ancestral bacterial symbionts [107–111], suggesting that fungi may establish more or less long-lasting relationships with insects and become an established part of their microbiota. Although fungal diversity was not explored in our study, it would likely deserve more attention in future work, and insects with a subterranean lifestyle are likely interesting candidates for such an investigation.

## Supporting information

**S1 Fig. Phylogenetic analysis of *Buchnera* associated with *A. corni* and its placement compared to strains associated with other aphid species, based on a sequence from the V3-V4 region of bacterial 16S rRNA.** The evolutionary history was inferred using the Neighbor Joining (NJ) methods, with a J-C model. The percentage of replicate trees was verified with bootstrap of 1000 replicates. Designations in bold are strains sequenced in this study. Host names are followed by the GenBank accession number of each bacterial sequence. Geneious version 6.1 created by Biomatters.
(TIF)

**S2 Fig. Phylogenetic analysis of *Serratia symbiotica* associated with *Anoecia corni* and their placement compared to strains associated with other aphid species, based on a sequence from the V3-V4 region of bacterial 16S rRNA.** The evolutionary history was inferred using the Neighbor Joining (NJ) methods, with a J-C model. The percentage of replicate trees was verified with bootstrap of 1000 replicates. Designation in bold are is strains sequenced in this study. Host names are followed by the GenBank accession number of each bacterial sequence. Geneious version 6.1 created by Biomatters.
(TIF)

**S3 Fig. Phylogenetic analysis of *Wolbachia* associated with *Anoecia corni* and its placement compared to strains associated with other insect species, based on a sequence from the V3-V4 region of bacterial 16S rRNA.** The evolutionary history was inferred using the Neighbor Joining (NJ) methods, with a HKY model. The percentage of replicate trees was verified with bootstrap of 1000 replicates. Designation in bold is strain sequenced in this study. Host names are followed by the GenBank accession number of each bacterial sequence. Geneious version 6.1 created by Biomatters.
(TIF)

**S4 Fig. Phylogenetic analysis of *Arsenophonus* associated with *Anoecia corni* and its placement compared to strains associated with other insect species, based on a sequence from the V3-V4 region of bacterial 16S rRNA.** The evolutionary history was inferred using the Neighbor Joining (NJ) methods, with a HKY model. The percentage of replicate trees was

verified with bootstrap of 1000 replicates. Designation in bold is strain sequenced in this study. Host names are followed by the GenBank accession number of each bacterial sequence. Geneious version 6.1 created by Biomatters.
(TIF)

**S1 Table. Summary of collection details and 16S rRNA gene sequencing results for aphid samples.**
(DOCX)

**S2 Table. Data analysis: Commands and options used to build the OUT table.**
(DOCX)

**S3 Table. Bacterial representative sequences (operational taxonomic units) found in *A. corni*.**
(DOCX)

**S4 Table. Contaminant OTUs identified in this study.**
(DOCX)

## Acknowledgments

The authors thank Linda Dhondt, Karim Andich and Slimane Khayi for technical assistance. This paper is publication BRC359 of the Biodiversity Research center (UC Louvain).

## Author Contributions

**Conceptualization:** Samir Fakhour, François Renoz, Thierry Hance.

**Data curation:** Samir Fakhour, François Renoz, Jérôme Ambroise.

**Formal analysis:** Samir Fakhour, François Renoz, Jérôme Ambroise, Inès Pons, Christine Noël, Jean-Luc Gala.

**Funding acquisition:** Samir Fakhour, Inès Pons, Thierry Hance.

**Investigation:** Samir Fakhour, Thierry Hance.

**Methodology:** Samir Fakhour, François Renoz, Thierry Hance.

**Project administration:** Thierry Hance.

**Software:** Jérôme Ambroise, Jean-Luc Gala.

**Supervision:** Thierry Hance.

**Validation:** Samir Fakhour, François Renoz, Thierry Hance.

**Visualization:** Samir Fakhour, François Renoz.

**Writing – original draft:** Samir Fakhour, François Renoz, Inès Pons.

**Writing – review & editing:** Samir Fakhour, François Renoz, Jérôme Ambroise, Inès Pons, Jean-Luc Gala, Thierry Hance.

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
