## [Decision Letter · Decision Letter 0]

13 May 2021

PONE-D-20-40214

Insight into the Microbiome of the Subterranean Aphid Anoecia corni

PLOS ONE

Dear Dr. Fakhour,

Thank you for submitting your manuscript to PLOS ONE. After careful consideration, we feel that it has merit but does not fully meet PLOS ONE’s publication criteria as it currently stands. Therefore, we invite you to submit a revised version of the manuscript that addresses the points raised during the review process.

I am really sorry for the long time that all this review process took; your manuscript has been evaluated by three independent reviewer and although one has recommended rejection, the other two say it has great value and should be published after revisions. So I kindly ask you to reply to all the reviewers and try to integrate their comments and suggestions in your revised manuscript

We look forward to receiving your revised manuscript.

Kind regards,

Clara F. Rodrigues

Academic Editor

PLOS ONE

Journal Requirements:

2. We note that Figure 1 in your submission contain map images which may be copyrighted. All PLOS content is published under the Creative Commons Attribution License (CC BY 4.0), which means that the manuscript, images, and Supporting Information files will be freely available online, and any third party is permitted to access, download, copy, distribute, and use these materials in any way, even commercially, with proper attribution. For these reasons, we cannot publish previously copyrighted maps or satellite images created using proprietary data, such as Google software (Google Maps, Street View, and Earth). For more information, see our copyright guidelines: http://journals.plos.org/plosone/s/licenses-and-copyright.

2.1.    You may seek permission from the original copyright holder of Figure(s) [#] to publish the content specifically under the CC BY 4.0 license. 

2.2.    If you are unable to obtain permission from the original copyright holder to publish these figures under the CC BY 4.0 license or if the copyright holder’s requirements are incompatible with the CC BY 4.0 license, please either i) remove the figure or ii) supply a replacement figure that complies with the CC BY 4.0 license. Please check copyright information on all replacement figures and update the figure caption with source information. If applicable, please specify in the figure caption text when a figure is similar but not identical to the original image and is therefore for illustrative purposes only.

Reviewers' comments:

Reviewer's Responses to Questions

**Comments to the Author**

1. Is the manuscript technically sound, and do the data support the conclusions?

Reviewer #1: Yes

Reviewer #2: Partly

Reviewer #3: Yes

2. Has the statistical analysis been performed appropriately and rigorously? 

Reviewer #1: Yes

Reviewer #2: Yes

Reviewer #3: N/A

3. Have the authors made all data underlying the findings in their manuscript fully available?

Reviewer #1: Yes

Reviewer #2: Yes

Reviewer #3: Yes

4. Is the manuscript presented in an intelligible fashion and written in standard English?

Reviewer #1: Yes

Reviewer #2: Yes

Reviewer #3: Yes

5. Review Comments to the Author

Reviewer #1: The manuscript by Fakhour and colleagues describes the bacterial microbiome associated with 16 samples of the aphid Anoecia corni collected on wheat roots in two regions of Morocco.

The manuscript is well written and presented in general, but it has limited interest even to those who specifically study insect microbiomes. The bacterial species identified in this work were already known to be associated with aphids, so the manuscript adds little to the current understanding about the aphid microbiome. As a general comment, the in silico study of microorganisms other than bacteria (e.g. fungi, which are key components of the plant rhizosphere) would have increased the novelty and interest of the paper. Insect microbiomes, in fact, include also fungi, viruses, and protozoa. As A. corni lives in close association with soil and wheat roots, at least the fungal component of the insect microbiome should have been explored.

In addition, environmental parameters associated to each sampling site (e.g. precipitation, temperature, etc…) are not provided. Such data, for example, might have shed light to the possible impact of the environment on the biodiversity indexes, which, by the way, are totally missing.

Minor issues

The title: as the Authors analyzed only the bacterial component of the aphid microbiome, I suggest they modify the title accordingly.

Abstract line 28: the Authors claimed that the bacterial microbiome of aphids is “poorly diversified”, but later in the Introduction they stated that aphids are associated “with a wide range of bacterial symbionts”. Please explain this contradiction.

Abstract line 42: In this study the Authors took into consideration only the bacterial endosymbionts, so it is obvious that the bacterial diversity found in A. corni samples is limited to this part of the insect microbiome. Please rephrase.

Table 2: What does “PC reads.” stands for?

Introduction page 3: “B. aphidicola” and “A. corni” should be written without abbreviations as it is the first time they are cited in the manuscript. Check the spelling of “Hamiltonella defensa”

Page 4 line 90: the correct spelling is “dioecious”

Page 6 and Materials and Methods in general: if the procedures are already described in another paper, I suggest to cite the paper and keep the description short, providing only relevant modifications from the previous protocols/approaches, if there are any.

Page 11 line 214: OTU_6 (and not OTU 7) corresponds to Wolbachia (Table 2) and I assume that OTU 18 is actually OTU 8 and OTU 19 is OTU 9, is it correct?

Figure 1 improve resolution and associate the library codes to the sampling locations.

Supplementary figures: phylogenetic trees should first report bacterial names and then their host names in brackets, otherwise they seem to represent insect phylogeny, instead of that of their symbionts. In some cases there are bacterial names and insect names on different branches of the same tree.

Reviewer #2: Overall comments :

In the present paper, the authors use 16S rRNA sequencing to investigate for the first time the microbial communities associated to the subterranean aphid Anocia comi.

They identified a total of 23 OTUs, 10 of them being associated to 5 putative symbiont genera, including the obligatory symbiont Buchnera aphipicola and 4 known facultative symbionts. Among them was Wolbachia, which is considered rare in aphids.

Overall, I am pleased by the results presented in the paper. However, some elements are unclear and need to be reworked to make the paper easier to understand and reproduce for other researchers. In particular, figures need to be reworked, and it seems (to me at least) that there is some confusion in sample naming and OTU numbering.

Hereafter are my comments. Most of them are typos or minor comment but some of them are more important for the comprehension of the paper.

Abstract :

37 : I'd remove "hosting"

42 : I don't understand this sentence. What is expected apart from bacterial endosymbionts?

Data analysis : for the sake of reproducibility, the commands and options used for the different software should be supplied.

161 : missing citation for dada2

167 : There is no correpondance given between the identifiers you used for your samples and the ENA identifiers. Could you please add the accessions in sup Table 1 for instance?

169 : missing citation for SeaView

Table 1 : Maybe add median sequence length after filtering?

More detailled statstics (min-median-max) would also be useful

186 : 0.23% is not much but is massive compared to the abundance of Wolbachia for instance. I would at least remove the "only".

Also, the complete abundance table for the 23 OTUs should be available as supplementary.

188 : The contaminant OTUs are also "genuine". I would use "biologically relevant"

Table 2 :

OTU identifiers do not match with the figures.

I'd appreciate a comment on the discrepancies between the Greengenes identification (from dada2?) and the Blast based

Fig 2 :

- The naming of samples is very unclear. I understand (from the supplementary) that the number indicates the locality. But this naming convention gives the impression that samples are paired. There is also a H12 sample which is puzzling to me.

- Could also indicate on Fig2 the geographic origin of samples?

- Text should be bigger (also for other figures)

- Apart from the presence absence of Serratia, the Figure is hard to read. Maybe use a log scale? And it's also quite redundant with Fig. 3

Fig 3 :

Legend : How is that "relative abundance"? It seems to me that these are absolute counts. The Figure with relative abundances would be Fig. 2.

I would rephrase "Each color bar corresponding" by "Each column represents"

You mention log counts, but the counts on the scale do not seem log transformed. I believe only the scale is logarithmic.

Maybe a heatmap made only of secondary symbions in addition to this one would be useful to understand where they are (ideally with read counts or relative abundances written on the heatmap)

217 - Fig S1-S4 : Overall the trees are poorly supported and not discussed in the paper.

234 : "Extremely low number of reads" : Please give numbers, in absolute (read counts) and relative (%).

The count table for all OTUs and all samples should be available as supplementary.

You make no comment on the fact that the different Buchnera OTUs are not evenly distributed across samples.

249 : Could it be because you used a different extraction protocol?

Reviewer #3: PLOS ONE

Review: “Insight into the Microbiome of the Subterranean Aphid Anoecia corni”

Authors: Samir Fakhour, François Renoz, Jérôme Ambroise, Inès Pons, Christine Noël, Jean- Luc Gala, Thierry Hance

General Comments

The study has an interesting premise to investigate the microbiota of ground dwelling, root feeding aphids. The paper provides a strong hypothesis as to why these aphids might harbor more microbes, and sets out to examine this using High Throughput Sequencing. The incorporation of phylogenetic relatedness of microbial organisms to support whether they are aphid, soil, or plant associated is good, although this concept should be mentioned in the introduction.

Overall this manuscript provides a good discussion of the 5 main bacterial OTUs found in A. corni. An overview of the functions these microbes in other organisms was provided along with the bacteria’s potential role in this aphid. The conclusion provided a nice summary of additional lines of investigation for the role of Arsenophonus and Wolbachia in A. corni.

Major Comments

Flow and organization can be enhanced

-Some sentences in the discussion have repetitive content and could be enhanced by restructuring or merging two sentences into one. Reminders throughout the text of the main ideas are helpful, however the sentence repetition does not help drive the manuscript content forward.

-Incorporate relevance into the introduction

-The introduction should mention the importance of phylogenetic relatedness in determining microbial associations, to prime the reader for the Neighbor Joining tree analyses in the materials and methods.

Clarify Materials and methods

-Based upon the sample collection information, it is not entirely clear the process for obtaining aphids and then raising the colonies. Were apterous adult aphids collected on wheat roots, then placed into colonies and then used to generate clones? Then were these aphid clones used to assess microbial diversity? It is unclear.

-Revise discussion for clarity and flow

-The manuscript makes a bold claim that no environmental bacteria were found associated with aphids, however only the aphids were sampled and not the plant or xylem fluids. The way that this is written should be framed in the context that bacterial symbionts were not more abundant in ground dwelling aphids despite being in contact with soil microbes and having access to xylem. An average or assessment of microbes identified in other aphids beyond the typical symbionts should be provided if this comparison is to be made.

-While after reading L239 to L234 it is clearer to the reader that the phylogenetic assessment using Neighbor Joining (NJ) trees were intended to clarify the relationship of microbial organisms to aphid hosts, the relevance of phylogenetic relationships among microbes and their environment/host association should be mentioned in the introduction to prep the reader for this in the materials and methods and then discussion.

Incorporate Citations

-Lacks some citations for aphid studies in the past 5 years that used HTS to examine microbiota composition and/or phylogenetic relationships of microbes. While the citations used are fine, the predominant use of older/foundational references is noticeable.

Minor Comments

L62: Update italicization from genus and species being italicized to Candidatus being italicized and genus and species being unitalicized

L81: Suggested update “microbiota associate” to “microbiota associated”

L91: Suggested update to “many whose ecological and taxonomic position remain largely unknown”

L101: change “microbiome” to “microbiota”

L126: change “step” to “steps”

L186: suggested to add the 3 OTUs that were removed as negative controls. “were identified as contaminants and removed (e.g. contaminant 1, contaminant 2, contaminant 3).”

L247: Dactylopiibacterium symbionts are noted to be found in scale insects (Vera-Ponce de León et al. Genome Biol Evol. 2017 and Bustamante-Brito et al. Life. 2019.). Where is the citation to support that this is a known associate of aphids? This sentence should have some citations provided.

L249: Revise to something along the lines of “We did not find any bacterial partners that can be considered as environmental-related (e.g. Pseudomonas spp., Erwinia spp., etc.) as in the case of other aphid species, including those that feed on cereal crops [29, 31].”

L252 – L253: Recommendation to combine both sentences into one: “S. symbiotica is one of the most common symbiont species in aphid populations [55] and was identified in three of the eight colonies surveyed.”

L259 – L261: Suggested reduction and clarification of sentences into one: “This symbiont, an α-proteobacterium, is commonly found in insects and studies suggest that Wolbachia is present in at least 65% of arthropod species [59].”

L300: Change “Quite few” to “Few”

References

Update references so that the first letter of the title is capitalized and the remaining portion of the title is in lower case. Also make sure to italicize scientific names of microbial organisms.

L354, L357, L470, L472, L475, L477, L485, L491, L498, L503, L508: Italicize Wolbachia

L371: Italicize Buchnera

L379: Remove all caps from manuscript title

L520, L526, L532: Italicize Arsenophonus

L543: Update italicization from genus and species being italicized to Candidatus being italicized and genus and species being unitalicized

Table 2

In Id% columns change the formatting from a “,” to a “.”. For example “98,28” would be formatted as “98.28”.

Figure 1

The map figure needs greater resolution as it is currently blurry. For someone unfamiliar with this area, the map is not very useful. It would be nice to have each point labeled with the colony name and then have this linked to the supplementary Table S1. GPS coordinates.

Figure 2

This figure is also blurry and needs greater resolution. Not all the colors labeled that correspond to different microbes are show in the graph. It would be helpful to clearly label the X and Y axis with what they represent. Also it is highly recommended to change the color of either Serratia or Wolbachia, as those with red-green colorblindness will be unable to distinguish between the two colors selected.

Figure 3

While the samples are labeled in figure 3, they are in the opposite orientation as those in Figure 2. This should be updated so that the sample text faces the same direction for both Fig 2 and 3. Also the figure resolution should be increased.

Table S2

Update font size in this supplementary document so that it is consistent for all OTUs, rather than size 10 font in some places and size 12 font in others.

Supplementary figures 1-4

-Greater resolution on figures needed.

-Neighbor joining trees have very low branch support values. In the context of this study, could be interpreted as distantly related bacterial strains still being affiliated with other insects. However, when it comes to Wolbachia from A. corni the low support (75%) from the neighbor joining tree and separation from all other strains of Wolbachia could be that this is a closely related bacteria masquerading as Wolbachia.

6. PLOS authors have the option to publish the peer review history of their article (what does this mean?). If published, this will include your full peer review and any attached files.

Reviewer #1: No

Reviewer #2: **Yes: **Cervin Guyomar

Reviewer #3: No

---

## [Author Response · Author response to Decision Letter 0]

26 Jul 2021

Please find attached the revised version of our manuscript "Insight into the Microbiome of the Subterranean Aphid Anoecia corni" (Manuscript [PONE-D-20-40214] - [EMID:8d559adb517b70dd]). We thank you and the reviewers for their relevant comments and encouragement. This reinforces the idea of intensifying our research efforts in using next generation sequencing approaches to decipher how bacterial communities (and beyond) are shaped in insect populations. All of the reviewers' comments have been addressed and the manuscript has been revised accordingly. Our responses are in red in this letter as well as in the new version of the manuscript. We hope that the changes and additions have sufficiently improved the manuscript to meet PLOS ONE standards. 

We would like to point out that, following the reviewers' comments, we have changed the title to "Insight into the bacterial communities of the Subterranean Aphid Anoecia corni".

We look forward to your response 

Sincerely, 

Samir Fakhour and co-authors

Dear Dr. Fakhour,

Thank you for submitting your manuscript to PLOS ONE. After careful consideration, we feel that it has merit but does not fully meet PLOS ONE’s publication criteria as it currently stands. Therefore, we invite you to submit a revised version of the manuscript that addresses the points raised during the review process.

I am really sorry for the long time that all this review process took; your manuscript has been evaluated by three independent reviewer and although one has recommended rejection, the other two say it has great value and should be published after revisions. So I kindly ask you to reply to all the reviewers and try to integrate their comments and suggestions in your revised manuscript

We look forward to receiving your revised manuscript.

Kind regards,

Clara F. Rodrigues

Academic Editor

PLOS ONE

Journal Requirements:

2. We note that Figure 1 in your submission contain map images which may be copyrighted. All PLOS content is published under the Creative Commons Attribution License (CC BY 4.0), which means that the manuscript, images, and Supporting Information files will be freely available online, and any third party is permitted to access, download, copy, distribute, and use these materials in any way, even commercially, with proper attribution. For these reasons, we cannot publish previously copyrighted maps or satellite images created using proprietary data, such as Google software (Google Maps, Street View, and Earth). For more information, see our copyright guidelines: http://journals.plos.org/plosone/s/licenses-and-copyright.

2.1. You may seek permission from the original copyright holder of Figure(s) [#] to publish the content specifically under the CC BY 4.0 license. 

2.2. If you are unable to obtain permission from the original copyright holder to publish these figures under the CC BY 4.0 license or if the copyright holder’s requirements are incompatible with the CC BY 4.0 license, please either i) remove the figure or ii) supply a replacement figure that complies with the CC BY 4.0 license. Please check copyright information on all replacement figures and update the figure caption with source information. If applicable, please specify in the figure caption text when a figure is similar but not identical to the original image and is therefore for illustrative purposes only.

Reply: Based on your suggestion, the correction has been made. The map has been changed to another figure with better resolution and the image of the map is not under copyright. The aphid colony code has been reported in this figure as shown in Table S1.

Answer: Based on your suggestion, the correction has been made.

Reviewers' comments:

Reviewer's Responses to Questions

Comments to the Author

1. Is the manuscript technically sound, and do the data support the conclusions?

Reviewer #1: Yes

Reviewer #2: Partly

Reviewer #3: Yes

2. Has the statistical analysis been performed appropriately and rigorously?

Reviewer #1: Yes

Reviewer #2: Yes

Reviewer #3: N/A

3. Have the authors made all data underlying the findings in their manuscript fully available?

Reviewer #1: Yes

Reviewer #2: Yes

Reviewer #3: Yes

4. Is the manuscript presented in an intelligible fashion and written in standard English?

Reviewer #1: Yes

Reviewer #2: Yes

Reviewer #3: Yes

5. Review Comments to the Author

Reviewer #1: The manuscript by Fakhour and colleagues describes the bacterial microbiome associated with 16 samples of the aphid Anoecia corni collected on wheat roots in two regions of Morocco.

The manuscript is well written and presented in general, but it has limited interest even to those who specifically study insect microbiomes. The bacterial species identified in this work were already known to be associated with aphids, so the manuscript adds little to the current understanding about the aphid microbiome. As a general comment, the in silico study of microorganisms other than bacteria (e.g. fungi, which are key components of the plant rhizosphere) would have increased the novelty and interest of the paper. Insect microbiomes, in fact, include also fungi, viruses, and protozoa. As A. corni lives in close association with soil and wheat roots, at least the fungal component of the insect microbiome should have been explored.

In addition, environmental parameters associated to each sampling site (e.g. precipitation, temperature, etc…) are not provided. Such data, for example, might have shed light to the possible impact of the environment on the biodiversity indexes, which, by the way, are totally missing.

We agree with these remarks and the limitations pointed out by the reviewer. This study was initially based on a large sampling campaign covering the cereal growing areas of Morocco and targeting cereal aphids and their associated bacterial communities. With this exploratory study, we had the ambition to report the bacterial diversity associated with an aphid receiving little attention and living in close contact with the soil microbial flora, with the hypothesis that they are potentially associated with a diverse bacterial community given their lifestyle. Obviously, this is not the case and the contrary would certainly have made the study more interesting (but the results are as they are...). The comment about identifying other types of microorganisms is also relevant, especially about fungal diversity, which was not tested here. It is true that this is an aspect that has been little explored in aphids and beyond, whereas recent studies have shown that in some species, the fungal diversity encountered could even be at the origin of new endosymbioses where bacteria are replaced by fungi. This is the case in some cicadas, grasshoppers, and aphids where fungi of the genus Ophicordyceps have become obligate symbionts by replacing ancestral bacterial symbionts (Suh, Noda and Blackwell 2001; Vogel and Moran 2013; Xue et al. 2014; Blackwell 2017; Matsuura et al. 2018). Insect-associated bacterial communities have received much attention over the past two decades, somewhat eliding fungal diversity that would certainly deserve more attention, and aphids having a subterranean lifestyle might be good models to begin such an exploration. Although this could not be done in our study, this perspective is now mentioned in the manuscript. 

L308-316: We added perspectives on the importance of assessing fungal diversity in insects.

Minor issues

The title: as the Authors analyzed only the bacterial component of the aphid microbiome, I suggest they modify the title accordingly.

Reply: we changed the title which is now: “Insight into the bacterial communities of the Subterranean Aphid Anoecia corni” (L1).

Abstract line 28: the Authors claimed that the bacterial microbiome of aphids is “poorly diversified”, but later in the Introduction they stated that aphids are associated “with a wide range of bacterial symbionts”. Please explain this contradiction.

Reply: Although there is ambiguity, there is no contradiction. Indeed, aphids are known to be associated with low bacterial diversity compared to other insects such as fruit flies, ants, termites, etc. which may be associated with a high diversity of environmental bacteria residing in their digestive tract. In aphids, this low bacterial diversity in the digestive tract stems from the fact that the phloem sap on which they feed is known to be virtually devoid of microorganisms (Grenier, Nardon and Rahbé 1994). Thus, aphid-associated bacterial communities are largely dominated by a few symbiont taxa (there are about a dozen, but an aphid rarely harbors more than three) as in other sap-feeding insects (Colman, Toolson and Takacs-Vesbach 2012; Jing et al. 2014). Thus, aphids are characterized as being associated with a low diversity bacterial community, which essentially revolves around a dozen species of bacterial symbionts. If the bacterial community associated with the aphid is low, on the other hand, the symbiont diversity is rather broad since most insects do not harbor heritable bacteria (at the exception of Wolbachia) but do harbor a large diversity of transient bacteria.

L53: We added the term "heritable" which specifies more precisely what we mean by symbiont (there is a durable and heritable character in the relationship maintained with the insect).

Abstract line 42: In this study the Authors took into consideration only the bacterial endosymbionts, so it is obvious that the bacterial diversity found in A. corni samples is limited to this part of the insect microbiome. Please rephrase.

Reply: We do not understand this comment. In this study, the 16S rRNA amplicon Illumina sequencing approach was used to assess the composition of the overall bacterial communities associated with our sampled aphids. It is true that the presence of other bacteria, for example living in the digestive tract, is deemed to be rare in aphids, as the phloem sap is considered an almost sterile environment (Grenier, Nardon and Rahbé 1994). Nevertheless, some studies have revealed the sporadic presence of extracellular bacteria have been in the digestive tract of aphids. These include beneficial gut symbionts, pathogens, plant pathogens, and environmental contaminants (Harada et al. 1997; Sevim, Çelebi and Sevim 2012; Gauthier et al. 2015). In a previous study on five species of cereal aphids, and in addition to the bacterial endosymbionts, bacteria from the genera Pseudomonas, Acinetobacter, Pantoea, Erwinia and Staphyloccocus were detected (Fakhour et al. 2018). Extracellular strains of the symbiont Serratia symbiotica could also be detected in the digestive tract (Pons et al. 2021). These observations thus suggest that the aphid microbiome is not necessarily limited to intracellular endosymbiotic bacteria alone. Therefore, our approach focused on all bacteria present in the sampled aphids. We did not consider only heritable symbionts, but we found almost only that.

However, to be more accurate, we have replaced in the abstract "symbiotic diversity" by "bacterial diversity" (L30). 

Table 2: What does “PC reads.” stands for?

Reply: PC reads corresponds to cluster size in percent. We have added a caption to line 204.

Introduction page 3: “B. aphidicola” and “A. corni” should be written without abbreviations as it is the first time they are cited in the manuscript. Check the spelling of “Hamiltonella defensa”

Reply: Based on your suggestion, the correction has been made (L56 & L62).

Page 4 line 90: the correct spelling is “dioecious”

Reply: The correction has been made (L90).

Page 6 and Materials and Methods in general: if the procedures are already described in another paper, I suggest to cite the paper and keep the description short, providing only relevant modifications from the previous protocols/approaches, if there are any.

Reply: We have shortened some procedures already mentioned in one of our previous studies (Fakhour et al. 2018). By rereading the manuscript, we confirm that there is nothing to change on this side.

Page 11 line 214: OTU_6 (and not OTU 7) corresponds to Wolbachia (Table 2) and I assume that OTU 18 is actually OTU 8 and OTU 19 is OTU 9, is it correct?

Reply: The correction has been made (L221 & L222).

Figure 1 improve resolution and associate the library codes to the sampling locations.

Reply: Based on your suggestion, the correction has been made.

Supplementary figures: phylogenetic trees should first report bacterial names and then their host names in brackets, otherwise they seem to represent insect phylogeny, instead of that of their symbionts. In some cases there are bacterial names and insect names on different branches of the same tree.

Reply: We agree with this comment. Phylogenetic trees have been modified according to your suggestion (Lines 668 to 698 and Fig S1, S2, S3 and S4).

Reviewer #2: Overall comments:

In the present paper, the authors use 16S rRNA sequencing to investigate for the first time the microbial communities associated to the subterranean aphid Anocia comi.

They identified a total of 23 OTUs, 10 of them being associated to 5 putative symbiont genera, including the obligatory symbiont Buchnera aphipicola and 4 known facultative symbionts. Among them was Wolbachia, which is considered rare in aphids.

Overall, I am pleased by the results presented in the paper. However, some elements are unclear and need to be reworked to make the paper easier to understand and reproduce for other researchers. In particular, figures need to be reworked, and it seems (to me at least) that there is some confusion in sample naming and OTU numbering.

Hereafter are my comments. Most of them are typos or minor comment but some of them are more important for the comprehension of the paper.

Abstract:

37 : I'd remove "hosting"

Reply: Based on your suggestion, the correction has been made (L37).

42 : I don't understand this sentence. What is expected apart from bacterial endosymbionts?

Reply: Bacteria residing in the digestive tract, pathogens, plant pathogens and environmental contaminants. Please, see comments above (reviewer 1) and the manuscript L67-71 for example. 

Data analysis: for the sake of reproducibility, the commands and options used for the different software should be supplied.

Reply: Based on your suggestion. We added the commands and options used for the different software in supporting information (S2 table. Data analysis: Commands and options used to build the OUT table).

161 : missing citation for dada2

Reply: The appropriate citation has been added (Callahan et al. 2016) (L163). 

167 : There is no correpondance given between the identifiers you used for your samples and the ENA identifiers. Could you please add the accessions in sup Table 1 for instance?

Reply: Based on your suggestion, the the ENA identifiers (sample accession ENA) has been added for each sample in S1.Table (second column of the table).

169 : missing citation for SeaView

Reply: The appropriate citation has been added (Gouy, Guindon and Gascuel 2010) (L172). 

Table 1 : Maybe add median sequence length after filtering?

More detailled statstics (min-median-max) would also be useful

Reply: Based on your suggestion, we added the Sequence length (min; median; max) in the Table 1 (page 8).

186 : 0.23% is not much but is massive compared to the abundance of Wolbachia for instance. I would at least remove the "only".

Also, the complete abundance table for the 23 OTUs should be available as supplementary.

Reply: We add a S4 Table. Contaminant OTUs identified in this study in the Supplementary files. That presents the 13 contaminant OTUs identified from negative control analysis with their size in percent (OTU 11 to 23).

188 : The contaminant OTUs are also "genuine". I would use "biologically relevant"

Reply: Based on your suggestion, the correction was made (L190).

Table 2 :

OTU identifiers do not match with the figures.

I'd appreciate a comment on the discrepancies between the Greengenes identification (from dada2?) and the Blast based

Reply: Greengenes databse is discontinued and not been updated since many years, also, there are several mistakes in the taxonomy (> 20%), so for that we have somes discrepancies.

Fig 2 :

- The naming of samples is very unclear. I understand (from the supplementary) that the number indicates the locality. But this naming convention gives the impression that samples are paired. There is also a H12 sample which is puzzling to me.

- Could also indicate on Fig2 the geographic origin of samples?

- Text should be bigger (also for other figures)

- Apart from the presence absence of Serratia, the Figure is hard to read. Maybe use a log scale? And it's also quite redundant with Fig. 3

Fig 3 : Legend : How is that "relative abundance"? It seems to me that these are absolute counts. The Figure with relative abundances would be Fig. 2.

I would rephrase "Each color bar corresponding" by "Each column represents"

You mention log counts, but the counts on the scale do not seem log transformed. I believe only the scale is logarithmic.

Maybe a heatmap made only of secondary symbions in addition to this one would be useful to understand where they are (ideally with read counts or relative abundances written on the heatmap)

Reply: Based on your suggestion, the correction was made. The Figure 2 has been removed to avoid redundancy. Figure 3 then becomes Fig. 2 and it has been improved according to your instructions.

217 - Fig S1-S4 : Overall the trees are poorly supported and not discussed in the paper.

Reply: We hope it's better now. However, we keep them as an indication. It is true that we do not develop these analyses and results. In our opinion, these trees are informative and should be left in the data supplements: they have been built on the basis of short sequences (unavoidable here with short 16S sequences of 450 bp) and are therefore not extremely robust (L222 to L224). 

234 : "Extremely low number of reads" : Please give numbers, in absolute (read counts) and relative (%). The count table for all OTUs and all samples should be available as supplementary.

Reply: Based on your suggestion, the correction was made. The tables S3 and S4 in the supplementary files have been improved and now present the numbers in absolute (read counts) and relative (%) for each OUT (representative and contaminants).

You make no comment on the fact that the different Buchnera OTUs are not evenly distributed across samples.

Reply: we have added details on the different haplotypes associated with aphids from the two regions of Morocco in results and discussion sections (Lines 209 to 218 and lines 258 to 262, respectively). 

249 : Could it be because you used a different extraction protocol?

Reply: On the contrary, we used the same extraction protocol as the one used in the previous study on cereal aphids (Fakhour et al. 2018). 

Reviewer #3: PLOS ONE

Review: “Insight into the Microbiome of the Subterranean Aphid Anoecia corni”

Authors: Samir Fakhour, François Renoz, Jérôme Ambroise, Inès Pons, Christine Noêl, Jean- Luc Gala, Thierry Hance

General Comments

The study has an interesting premise to investigate the microbiota of ground dwelling, root feeding aphids. The paper provides a strong hypothesis as to why these aphids might harbor more microbes, and sets out to examine this using High Throughput Sequencing. The incorporation of phylogenetic relatedness of microbial organisms to support whether they are aphid, soil, or plant associated is good, although this concept should be mentioned in the introduction.

Overall this manuscript provides a good discussion of the 5 main bacterial OTUs found in A. corni. An overview of the functions these microbes in other organisms was provided along with the bacteria’s potential role in this aphid. The conclusion provided a nice summary of additional lines of investigation for the role of Arsenophonus and Wolbachia in A. corni.

Major Comments

Flow and organization can be enhanced

-Some sentences in the discussion have repetitive content and could be enhanced by restructuring or merging two sentences into one. Reminders throughout the text of the main ideas are helpful, however the sentence repetition does not help drive the manuscript content forward.

Reply: We have reread the manuscript several times, but without more details from you, it is difficult to know exactly which part to change. The introduction is quite direct, short, without redundancy. Same thing for the discussion which is quite short. The only slight redundancy is in the discussion to remind the interest of these aphids (subterranean lifestyle) in the light of the overall results. So, if you think some syntax adjustments are necessary, please give us more details.

-Incorporate relevance into the introduction

Reply: Again: what do you mean? Could you clarify this?

-The introduction should mention the importance of phylogenetic relatedness in determining microbial associations, to prime the reader for the Neighbor Joining tree analyses in the materials and methods.

Reply: With all due respect, we prefer to leave things as they are. Indeed, reviewer 2 is not very convinced of the approach, which we understand. Phylogenetic trees are based on the alignment of rather short 16S DNA sequences (about 450 bp). This does not provide good robustness (probably at least 1500 bp and preferably several concatenated genes). The trees are indicative, but given their lack of robustness (despite appreciable node values with the new analyses) we think it is more appropriate to leave them in the data supplements section to show that one is cautious and not to start speculating on these results (probably with longer reads and more genes the topologies would have been different).

Clarify Materials and methods

-Based upon the sample collection information, it is not entirely clear the process for obtaining aphids and then raising the colonies. Were apterous adult aphids collected on wheat roots, then placed into colonies and then used to generate clones? Then were these aphid clones used to assess microbial diversity? It is unclear.

Reply: In fact, we do not understand where this is not clear. We went to the field and directly collected the insects which were directly placed in tubes filled with ethanol. We did not generate clones in the laboratory. What we did was pretty standard (Fakhour et al. 2018).

-Revise discussion for clarity and flow

Reply: Once again, we have re-read the discussion. We feel that it is short, fluid and the sentences are short. If sections of the discussion are not well written or obscure, please give us more information so we can do the best job possible. 

-The manuscript makes a bold claim that no environmental bacteria were found associated with aphids, however only the aphids were sampled and not the plant or xylem fluids. The way that this is written should be framed in the context that bacterial symbionts were not more abundant in ground dwelling aphids despite being in contact with soil microbes and having access to xylem. An average or assessment of microbes identified in other aphids beyond the typical symbionts should be provided if this comparison is to be made.

Reply: We hope it's better now (L307-314). 

-While after reading L239 to L234 it is clearer to the reader that the phylogenetic assessment using Neighbor Joining (NJ) trees were intended to clarify the relationship of microbial organisms to aphid hosts, the relevance of phylogenetic relationships among microbes and their environment/host association should be mentioned in the introduction to prep the reader for this in the materials and methods and then discussion.

Reply: We hope it's better now (L102-103). 

Incorporate Citations

-Lacks some citations for aphid studies in the past 5 years that used HTS to examine microbiota composition and/or phylogenetic relationships of microbes. While the citations used are fine, the predominant use of older/foundational references is noticeable.

Reply: We have updated some citations for aphid studies in the past 5 years. 

- We add : Tian et al., 2019 (L61) ; Skaljac et al., 2018 (L62) ; Kaech and Vorburger, 2020 (L62) ; Polin et al., 2015 (L62) ; Romanov et al., 2020 (L63) ;

- The Jamin et al, 2019 and Parker et al., 2021 citations have replaced Scarborough et al., 2005 (L61)

- The Moreira et al., 2019 and Ren et al., 2020 citations have replaced Augustinos et al., 2011 and Gomez-Valero et al., 2004 (L63)

- The Mathé-Hubert et al., 2019 and Guidolin et al., 2018 citations have replaced Fukatsu et al., 2001(L64)

- The Leclair et al., 2021; Chevignon et al., 2018 ; Oliver and Higashi, 2018, Heyworth and Ferrari, 2015 and Frago et al., 2017 citations have replaced Oliver et al., 2003 and Scarborough et al., 2005 (L65)

- The Tsuchida, 2016 and Nikoh et al., 2018 citations have replaced Tsuchidaet al., Tsuchida et al., 2004 (L65)

- The Heyworth et al., 2020 citation replaced Burke et al., 2010 (L66)

- The Lenhart & White, 2020 and Wagner et al., 2015 citations have replaced Tsuchida et al., 2004and Koga et al., 2003 (L66)

Minor Comments

L62: Update italicization from genus and species being italicized to Candidatus being italicized and genus and species being unitalicized

Reply: Based on your suggestion, the correction has been made (L62).

L81: Suggested update “microbiota associate” to “microbiota associated”

Reply: Based on your suggestion, the correction has been made (L81).

L91: Suggested update to “many whose ecological and taxonomic position remain largely unknown”

Reply: Based on your suggestion, the correction has been made (L91-92).

L101: change “microbiome” to “microbiota”

Reply: Based on your suggestion, the correction has been made (L101).

L126: change “step” to “steps”

Reply: Based on your suggestion, the correction has been made (L128).

L186: suggested to add the 3 OTUs that were removed as negative controls. “were identified as contaminants and removed (e.g. contaminant 1, contaminant 2, contaminant 3).”

Reply: Based on your suggestion, we added some information’s concerning de 13 (not 3) OTU removed as negative controls in the tables S4 Table. Contaminant OTUs identified in this study (see supplementary files).

L247: Dactylopiibacterium symbionts are noted to be found in scale insects (Vera-Ponce de León et al. Genome Biol Evol. 2017 and Bustamante-Brito et al. Life. 2019.). Where is the citation to support that this is a known associate of aphids? This sentence should have some citations provided.

Reply: At present, this bacterium has only been identified in these insects. It is thus for the first time (and it is a bit curious) that this species is reported in aphids. We cannot therefore include references related to aphids for this bacterium...

L249: Revise to something along the lines of “We did not find any bacterial partners that can be considered as environmental-related (e.g. Pseudomonas spp., Erwinia spp., etc.) as in the case of other aphid species, including those that feed on cereal crops [29, 31].”

Reply: Based on your suggestion, the correction has been made (L255-257).

L252 – L253: Recommendation to combine both sentences into one: “S. symbiotica is one of the most common symbiont species in aphid populations [55] and was identified in three of the eight colonies surveyed.”

Reply: Based on your suggestion, the correction has been made (L263-264).

L259 – L261: Suggested reduction and clarification of sentences into one: “This symbiont, an α-proteobacterium, is commonly found in insects and studies suggest that Wolbachia is present in at least 65% of arthropod species [59].”

Reply: Based on your suggestion, the correction has been made (L270-272).

L300: Change “Quite few” to “Few”

Reply: Based on your suggestion, the correction has been made (L318).

References

Update references so that the first letter of the title is capitalized and the remaining portion of the title is in lower case. Also make sure to italicize scientific names of microbial organisms.

L354, L357, L470, L472, L475, L477, L485, L491, L498, L503, L508: Italicize Wolbachia

L371: Italicize Buchnera

L379: Remove all caps from manuscript title

L520, L526, L532: Italicize Arsenophonus

Reply: Based on your suggestions, all requested changes have been made.

L543: Update italicization from genus and species being italicized to Candidatus being italicized and genus and species being unitalicized

Reply: Based on your suggestion, the correction has been made. 

Table 2

In Id% columns change the formatting from a “,” to a “.”. For example “98,28” would be formatted as “98.28”.

Reply: These changes have been made. 

Figure 1

The map figure needs greater resolution as it is currently blurry. For someone unfamiliar with this area, the map is not very useful. It would be nice to have each point labeled with the colony name and then have this linked to the supplementary Table S1. GPS coordinates.

Reply: We have made the requested changes. We hope that the new map is more suitable.

Figure 2

This figure is also blurry and needs greater resolution. Not all the colors labeled that correspond to different microbes are show in the graph. It would be helpful to clearly label the X and Y axis with what they represent. Also it is highly recommended to change the color of either Serratia or Wolbachia, as those with red-green colorblindness will be unable to distinguish between the two colors selected.

Figure 3

While the samples are labeled in figure 3, they are in the opposite orientation as those in Figure 2. This should be updated so that the sample text faces the same direction for both Fig 2 and 3. Also the figure resolution should be increased.

Reply: Based on your suggestions of those the reviewer 2, the figure 2 has been removed to avoid redundancy. Figure 3 then becomes Fig. 2 and it has been improved according to your instructions.

Table S2

Update font size in this supplementary document so that it is consistent for all OTUs, rather than size 10 font in some places and size 12 font in others.

Reply: Based on your suggestion, the correction has been made.

Supplementary figures 1-4

-Greater resolution on figures needed.

Reply: Based on your suggestion, the correction has been made.

-Neighbor joining trees have very low branch support values. In the context of this study, could be interpreted as distantly related bacterial strains still being affiliated with other insects. However, when it comes to Wolbachia from A. corni the low support (75%) from the neighbor joining tree and separation from all other strains of Wolbachia could be that this is a closely related bacteria masquerading as Wolbachia.

Reply: We have improved the branch support values of the phylogenetic trees (Figure S1-4). However, the analyzes were performed on the basis of short sequences (16S sequences of 450bp), which may explain these weak supports.

6. PLOS authors have the option to publish the peer review history of their article (what does this mean?). If published, this will include your full peer review and any attached files.

Do you want your identity to be public for this peer review? For information about this choice, including consent withdrawal, please see our Privacy Policy.

Reviewer #1: No

Reviewer #2: Yes: Cervin Guyomar

Reviewer #3: No

---

## [Editor Report · Decision Letter 1]

29 Jul 2021

Insight into the bacterial communities of the Subterranean Aphid Anoecia corni

PONE-D-20-40214R1

Dear Dr. Fakhour,

We’re pleased to inform you that your manuscript has been judged scientifically suitable for publication and will be formally accepted for publication once it meets all outstanding technical requirements.

Kind regards,

Clara F. Rodrigues

Academic Editor

PLOS ONE

Additional Editor Comments (optional):

Thank you for integrate all the comments and suggestions
---

## [Editor Report · Acceptance letter]

2 Aug 2021

PONE-D-20-40214R1 

Insight into the bacterial communities of the Subterranean Aphid *Anoecia corni*

Dear Dr. Fakhour:

I'm pleased to inform you that your manuscript has been deemed suitable for publication in PLOS ONE. Congratulations! Your manuscript is now with our production department. 

Kind regards, 

on behalf of

Dr. Clara F. Rodrigues 

Academic Editor

PLOS ONE